# Preparation of Berberine@carbon Dots Nano-Formulation: Synthesis, Characterization and Herbicidal Activity against *Echinochloa crus-galli* and *Amaranthus retroflexus* Two Common Species of Weed

**DOI:** 10.3390/nano12244482

**Published:** 2022-12-18

**Authors:** Junhu Xu, Shuang Rong, Ziqi Qin, Guangmao Shen, Yan Wu, Zan Zhang, Kun Qian

**Affiliations:** 1College of Plant Protection, Southwest University, Chongqing 400715, China; 2The National Center for Nanoscience and Technology, Beijing 100190, China

**Keywords:** berberine, CDs, *Echinochloa crus-galli*, herbicidal activity, soil colonies

## Abstract

Berberine (Ber) is easy to synthesize and has a variety of biological and pharmacological activities. At present, the existing studies on berberine have focused predominantly on its antibacterial activity; its herbicidal activity is rarely reported. In addition, there are a number of preparations of berberine, which are not enough to solve its shortcomings of low solubility and biological activity and the difficult storage of berberine. Here, berberine was combined with carbon dots to obtain carbon dots-berberine (CDs-Ber) nano formulation. The fluorescence quenching results showed that the CDs-Ber nano drug delivery system was successfully constructed, and the fluorescence quenching mechanism of the two was static quenching. The bioassay results showed that CDs had no adverse effects on the growth of barnyard grass (*Echinochloa crus-galli*) and redroot pigweed (*Amaranthus retroflexus*), and had high biocompatibility. Berberine and CDs-Ber predominantly affected the root growth of barnyard grass and redroot pigweed and could enhance the growth inhibition effect on weeds, to some extent. The results of the protective enzyme system showed that both berberine and CDs-Ber could increase the activities of Superoxide dismutase (SOD), Peroxidase (POD), and Catalase (CAT) in barnyard grass, and CDs-Ber had a stronger stress effect on barnyard grass than berberine. The determination of the number of bacterial communities in the soil after the berberine and CDs-Ber treatments showed that there was no significant difference in the effects of the two, indicating that CDs-Ber would not have more negative impacts on the environment. The CDs-Ber nano formulation improved the biological activity of berberine, enhanced the herbicidal effect, and was relatively safe for soil colonies.

## 1. Introduction

Pesticides play an important role in increasing food production and ensuring food security. Chemical pesticides account for more than 80% of China’s pesticide usage, with an average annual loss of approximately 40% of global crop production [1]. At the same time, the overdose of chemical pesticides has also caused food safety and environmental risks. With the deterioration of the environment, reducing the use of traditional chemicals has become an urgent problem to be solved.

Plant-derived pesticides are one of the preferred green pesticides. Berberine (Ber) is a kind of isoquinoline quaternary ammonium alkaloid with a variety of biological and pharmacological activities, which have been widely studied in the field of medicine [2]. In agriculture, many scholars have found that it had herbicidal [3,4], bactericidal [5], and insecticidal [6] activities in recent years. Due to its low solubility and biological activity, conventional formulations such as granules and powders [7], and nano formulations such as microspheres, microemulsions, and liposomes [8], have been developed. However, there are few commercial preparations of berberine, and there is no preparation that could solve the shortcomings of its low solubility and bioactivity and the difficult storage of berberine. Therefore, it is necessary to further explore and develop new berberine formulations. CDs are a new type of small-size nanomaterials with excellent optical properties, which were first reported in 2004 [9]. Compared with the traditional semiconductor quantum dots, CDs have the advantages of a small particle size, good dispersion, low environmental pollution, and high biocompatibility, and its many functional groups on the surface could lead it to be further modified. They are widely used in drug delivery therapy [10,11], biological imaging [12,13], photocatalysis [14,15], and biological detection [12,16]. In agriculture, CDs have great application potential in promoting plant growth and development, improving photosynthesis, and enhancing stress resistance and antibacterial activity [17,18]. In addition, the unique and stable fluorescence properties of CDs provide a new way to reveal their absorption and transport in plants [19]. In the application of carriers, the hydrophobic structure of CDs possess a hydrophobic interaction with other aromatic molecules, and the functional groups at the edge are easy to covalently bind to, or electrostatically interact with, other compounds. This could not only increase the stability of drugs and improve the solubility of hydrophobic molecules in water, but also transmit the drug aggregation to the target, reduce the concentration of non-target drugs, reduce toxicity, and improve efficiency [10,20]. At present, the research on CDs in drug carriers focuses predominantly on its medical applications and less in agriculture. Therefore, the application of CDs as drug carriers in agriculture is in urgent need of development.

Weeds indirectly affect crop production by competing with crops for resources, sheltering crop pests, interfering with water resources management, reducing yield and quality, and subsequently increasing the processing costs [21]. Barnyard grass (*Echinochloa crus-galli*) is one of the worst weed species in the world [22]. When large-scale barnyard grass infection occurs, it causes up to 80% nitrogen loss in the soil, and even becomes a secondary host of pathogens and viruses in some crops [23], which is fatal to the growth of farmland crops. Related studies have shown that barnyard grass is highly competitive in the growth of rice (*Oryza sativa* L.), resulting in 40–60% yield loss, even up to 90% if not properly controlled [23]. Redroot pigweed (*Amaranthus retroflexus*) is one of the most common annual broadleaf weeds in the world. It has a high seed yield, strong vigor, long germination period, and fast growth, which is difficult to manage in farmland [24]. When a large number of herbicides are used to control it, the red root grass produces three mechanisms of resistance: photosystem II (PSII), protoporphyrinogen oxidase (PPO), and ALS [25], making it increasingly difficult to control. In addition, the use of barnyard grass and red root grass as green feed and animal feeding is also fatal [26]. Although both species can be domesticated as food for human consumption, there are also negative effects of its uncontrolled growth and dispersal. Therefore, the development of new pesticide formulations is of great significance for controlling weeds and improving environmental safety.

Here, we prepared CDs loaded with a berberine nano formulation. The system was characterized by a transmission electron microscope, infrared spectrometer, and fluorescence spectrophotometer. The herbicidal activities of berberine and CDs-Ber against weeds were compared, and the safety of berberine and its preparations were evaluated in combination with their effects on the number of bacteria in the soil, which provided a theoretical basis for the new nano formulations and the application of berberine.

## 2. Materials and Methods

### 2.1. Materials Reagents and Seeds

Citric acid (CA, AR) was purchased from Sigma-Aldrich (Shanghai, China). Potassium bromide (SP), ethylenediamine (EDA, AR), and berberine (98%) were purchased from Maclin (Shanghai, China). Polyacrylic acid (PAA, BR) and anhydrous calcium chloride were purchased from Aladdin (Shanghai, China). Acetone (AR) and acetonitrile (AR) were purchased from Chuandong Chemical Group (Chongqing, China). Quinine sulfate (GBW) was purchased from the National Institute of Metrology, (China). Formic acid was purchased from TCL (Shanghai, China). Nutrient agar (NA), and potato dextrose agar (PDA) were purchased from Haibo Biotechnology (Qingdao, China). Barnyard grass (*Echinochloa crus-galli*) seeds and redroot pigweed (*Amaranthus retroflexus*) seeds were purchased from a Seed industry company (Anhui, China). Peroxidase (POD), Superoxide dismutase (SOD), and Catalase (CAT) Activity Assay Kit were purchased from Solarbio (Beijing, China).

### 2.2. Preparation and Characterization of PAA Modified CDs

The CDs were prepared from citric acid, ethylene diamine and polyacrylic acid, according to a published method [27]. A solution of 10 g CA and 6 mL EDA in 100 mL deionized water was stirred in three flasks for 4 h at 500 rpm at 200 °C. Next, 90 mL of 10% PAA solution was added, and the mixture was stirred for 3 h at 80 °C. We added three times the volume of acetone washed, centrifuged it at 10,000 r/min for 5 min, and discarded the supernatant. The lower solid was freeze-dried to obtain CDs, and stored at 4 °C.100 mg/L (Figure 1). The CDs solution was prepared, and its fluorescence phenomenon was observed under a 365 nm ultraviolet lamp (control was deionized water).

The morphology of the CDs was observed by transmission electron microscopy (TEM), and the particle size distribution data were counted (the solvent was deionized water, and the concentration was 1 mg/mL).

The structure and functional groups of CDs were characterized by Fourier transform infrared (FTIR) spectroscopy in the wavelength range of 400–4000 cm^−1.^ The sample was ground with KBr in a certain proportion.

The full spectrum of CDs was scanned by X-ray photoelectron spectroscopy (XPS), and the 3 elements of C, O, and N were accurately scanned.

The fluorescence spectra of CDs at different excitation and emission wavelengths were scanned by fluorescence spectrophotometer. The instrument sensitivity is 5, low-speed scanning. Solution concentration 100 mg/L, solvent, and deionized water.

### 2.3. Preparation and Characterization of CDs-Ber

The CDs solution (100 mg/L) and 200 mg/L berberine solution were mixed in equal volume and stirred for 12 h at 25 °C. The mixed solution was filtered through a filter membrane and dialyzed for 12 h with deionized water (refresh the water every 2 h). The CDs-Ber was obtained by vacuum freeze drying. The 0–100 mg/L berberine solution was prepared and its full spectrum was scanned by fluorescence spectrophotometer. The UV-Vis absorption spectra of berberine were detected by ultraviolet spectrophotometer.

The berberine solution of 0–200 mg/L was prepared and loaded into the CDs solution of 100 mg/L. The fluorescence intensity and fluorescence lifetime were detected by fluorescence spectrophotometer.

### 2.4. Herbicidal Activity of Berberine and CDs-Ber

Seeds of barnyard grass and redroot pigweed were soaked in 10% sodium hypochlorite solution and transferred to a petri dish covered with cotton and gauze for 24 h in darkness to bud. A total of 15 weed seeds with uniform size and uniform whiteness were selected and placed into petri dishes laid with qualitative filter paper, 9 mL of 15.6, 31.25, 62.5, 125, 250, and 500 mg/L berberine solution was added (each treatment was repeated 4 times, and deionized water was used as a blank control). The processed petri dishes were placed in a greenhouse at 28 °C for 3 days under dark conditions, and then cultured with a photoperiod of 14 L: 10 D for 4 days. The root length of the weeds was measured, and the inhibition rate was calculated after 7 days. The regression equation of toxicity was established and the median inhibitory concentration of berberine on the root length of the weeds was obtained.

Berberine, CDs, and CDs-Ber solution (CDs-Ber is the active ingredient content of berberine, the same as below) were prepared. The concentration was the IC50 concentration obtained by the regression equation, and deionized water was used as a control. The above plate method was used for the test. After 7 days, the root length and plant height of the weeds were measured, and the average value was calculated.

The indoor efficacy tests of two kinds of herbicide were carried out using the hydroponic method [28]. Sponge discs of uniform size were soaked and placed in a plastic cup filled with water. The seeds of barnyard grass were selected and placed into the sponge and cultured for 15 days at 25 ± 1 °C with a photoperiod of 14 L:10 D. These were subsequently transferred into plastic cups containing the berberine solution and CDs-Ber solution of the same volume and different concentrations, repeated 4 times. After 5 days, the damage symptoms of the barnyard grass were observed, and the root length and plant height were measured.

### 2.5. Effect of Berberine and CDs-Ber on Protective Enzyme System

The root tissue and leaf tissue of 0.1 g barnyard grass were respectively taken into the centrifuge tube, pre-cooled in liquid nitrogen for 30 min, and 1 mL buffer extraction solution and 1 steel ball were added. The samples were ground in a ball mill until the tissues were homogenized, and were then centrifuged in a low-temperature centrifuge at 4 °C and 8000 r/min for 15 min. The obtained supernatant was the crude enzyme extract of the barnyard grass. The Peroxidase (POD), Superoxide dismutase (SOD), and Catalase (CAT) enzyme activities were detected using kits. Each treatment was repeated 3 times.

### 2.6. Effects of Berberine and CDs-Ber on Soil Microbial Population

The microflora of the treated soil was counted using the plate counting method. The dry soil was weighed and added into the conical flask. The same volume of deionized water, berberine solution, and CDs-Ber solution were added, shaken in the shaker at 28 °C and 250 r/min for 30 min and stood for 15 min. The upper soil suspension was diluted with sterile water and 5–10 sterilized glass beads were added to the PDA medium and NA medium. Subsequently, 100 μL of bacterial fluid was added to each medium and shaken slowly, from left to right, until the bacterial fluid was slightly dry to ensure it can be evenly distributed on the medium (Figure 1). The processed plate seal was placed in the incubator at 28 °C and cultured upside down for 2 days. The number of single colonies was observed and recorded. Each treatment was repeated 4 times.

### 2.7. Statistical Analysis

Microsoft Excel 2016 was used for the data statistics, and Origin 2019b was used for the curve fitting and plotting. The virulence regression equation and analysis were performed using the SPSS 18.0 software, and the GraphPad Prism 8 software was used for plotting.

## 3. Result and Discussion

### 3.1. Morphology and Properties of CDs

Figure 1a,b shows the appearance of water and CDs under sunlight and UV lamps, respectively. Using citric acid and ethylenediamine as raw materials, the CDs were prepared with a dark green appearance and strong blue emission fluorescence. The morphology and particle size were characterized by transmission electron microscopy. Figure 1c,d indicates that the prepared CDs were evenly dispersed and approximately spherical. The particle size was mainly in the range of 2.5~20.0 nm, and the average particle size was 8.45 nm. This is essentially in line with previous reports [29].

In order to explore the structure and surface of the functional groups of the CDs sample, the elemental composition of the CDs samples was analyzed by XPS. As shown in Figure 2a, the XPS spectra show three peaks at 532.1 eV, 401.1 eV, and 285.1 eV, corresponding to the characteristic binding energies of C_1S_, N_1S_, and O_1S_, respectively. This indicates that the CDs mainly contain carbon, oxygen, and nitrogen [30] at contents of 60.16%, 30.58%, and 9.26%, respectively. As can be seen from Figure 2b, the C_1S_ spectrum of these CDs has three peaks at 288.5 eV, 286.3 eV, and 284.7 eV, corresponding to C=O [30], C-O [31] and C-C [32], respectively. The characteristic binding energies of C-O and C=O [33], corresponding to 532.5 eV and 531.6 eV, can be seen in the XPS spectrum of O_1S_ in Figure 2c. As shown in Figure 2d, the nitrogen elements in these CDs can be divided into two categories; namely, pyrrole nitrogen (C-N-C) at 401.4 eV and pyridine nitrogen (N-H) at 399.8 eV [30,32].

The results from the infrared spectrometer are shown in Figure 2e. The stretching vibration characteristic absorption peaks of -C=O at 1708 cm^−1^ and 1396 cm^−1^ indicate that the surface of the CDs contains carboxyl functional groups. The stretching vibration and bending vibration characteristic absorption peaks of -OH at 3417 cm^−1^ and 1057 cm^−1^ indicate that the surface of the CDs has a large number of hydroxyl groups in addition to carboxyl groups. The -C-N stretching vibration peak at 1572 cm^−1^ represents the formation of a -CONR bond, indicating that the surface of the CDs contain -NH_2_ groups. This result is essentially consistent with previous studies [34,35]. The above results confirm that there are functional groups, such as carboxyl, hydroxyl, and amino, on the surface of the CDs. These functional groups, on one hand, can be used as hydrophilic groups to endow CDs with good water solubility; on the other hand, they can provide abundant surface modification sites for CDs. It also provides the basis for the modification of CDs and their use as a carrier.

### 3.2. Optical Properties of CDs

To explore the optical properties of CDs, a multifunctional microplate reader and fluorescence spectrophotometer were used to conduct the spectral scanning of the CDs. from the results in Figure 3a show that the maximum excitation wavelength of the CDs was 360 nm, the maximum emission wavelength was 455 nm, and there was a weak absorption peak at about 280 nm caused by the conjugate double bond π-π transition of the C=C skeleton [30]. The solution showed bright blue fluorescence when irradiated with a 365 nm UV lamp. To further study the optical properties of the CDs, the fluorescence changes in the CDs at a series of wavelengths were measured, and the results are shown in Figure 3b,c. The results show that the maximum emission wavelength of the CDs changes with the increase in the excitation wavelength, which possess an excitation-dependent emission property. This phenomenon is similar to the reported properties of CDs, presumably due to the uneven particle size or surface passivation of the carbon dots caused by different chemicals [36]. With the increase in the excitation wavelength, the fluorescence intensity of the CDs first increased and then decreased. When the excitation wavelength was greater than 370 nm, the fluorescence intensity of the carbon point gradually decreased, but the peak position of the emission wavelength remained at 455 nm. Then, the influence of changing the emission wavelength on the fluorescence intensity of the CDs was investigated, and the maximum emission wavelength of the CDs was determined to be 455 nm. Finally, our study found that when the excitation wavelength of 700–800 nm was used to irradiate the CDs, the position of its fluorescence emission peak did not change with the change of the excitation wavelength, showing the property of upconversion luminescence [37], as shown in Figure 3d.

The results show that CDs prepared by hydrothermal synthesis have small particle sizes, good dispersibility, rich surface functional groups, and excellent fluorescence performance, providing a theoretical basis for their subsequent use as a carrier for drug loading and delivery.

### 3.3. Characterization and Optimization of CDs-Ber Nanodrug Delivery System

Figure 4b,c shows the change in the fluorescence intensity of the CDs after the addition of berberine with different concentrations. The fluorescence intensity of the CDs decreased with the increase in the berberine concentration, and the intensity of the CDs’ fluorescence quenching had a good linear relationship with berberine concentration within a certain range (0–25 mg/L). The linear equation was F_0_-F = 3.0931C-3.2518, and the correlation coefficient was 0.9915. Figure 4a shows the berberine UV-Vis absorption spectra, the carbon excitation, and the emission spectra of berberine. Berberine had absorption peaks in visible light at 380 nm and 421 nm, respectively. The absorption spectra of the excitation and emission wavelengths of berberine and CDs overlapped a lot, which implies that the fluorescence of the CDs was weakened due to the inner filter effect (IFE) [38]. This indicates that berberine was successfully carried.

Fluorescence quenching is the phenomenon in which the luminosity decreases, the luminescence time decreases, or the luminescence stops due to the interaction of fluorescent molecules with other molecules, which includes both the dynamic quenching effect (DQE) and static quenching effect (SQE) [16]. The former is due to the loss of excitation energy after the fluorescent molecule collides with the quencher, while the latter is due to the fluorescent molecule combining with other substances to form a ground state complex that does not change its properties and no longer emits light [39]. The fluorescence lifetime curves of the CDs and CDs-Ber were almost identical when the fluorescence lifetime curves of CDs and CDs-Ber were studied (Figure 4d), indicating that berberine had no effect on the fluorescence lifetime of CDs. The fluorescence quenching mechanism was SQE, indicating that berberine was bound to CDs rather than mixed, indicating that CDs-Ber was successfully prepared.

### 3.4. Effects of Berberine and CDs-Ber on Weed Growth

Barnyard grass and redroot pigweed were selected as model weeds, and the herbicidal activity of berberine was determined using the plate method. The results of the experiment showed that with the increase in the berberine concentration, its inhibitory effect on the root length of barnyard grass and redroot pigweed was also significantly enhanced. The toxicity regression equation was obtained by fitting the logarithm of berberine concentration with the inhibition rate of the root length. The results are shown in Table 1, and the correlation coefficients were 0.993 and 0.987, with good linear relationships, respectively. The inhibitory concentration of berberine on barnyard grass was 30.48 mg/L and 47.52 mg/L on redroot pigweed, which indicates that the herbicidal activity of berberine on barnyard grass was better than that of redroot pigweed.

To explore whether berberine and CDs-Ber had different biological activities on weeds, the concentration of berberine inhibiting the growth of two weeds was selected as the research parameter, and the following studies were carried out. In Figure 5, it was found that, compared with the treatment group, CDs, berberine, and CDs-Ber had no significant inhibitory effect on the plant height of barnyard grass but had a significant inhibitory effect on the plant height of redroot pigweed. Berberine and CDs-Ber could significantly inhibit the root growth of barnyard grass and redroot pigweed; the CDs promoted the growth of barnyard grass and the inhibitory effect of CDs-Ber was stronger than that of berberine, which might be absorbed in the increase in the berberine uptake in the weed with the effect of CDs.

To summarize, CDs-Ber can enhance the growth inhibition effect on weeds to some extent. We speculated that CDs were small in volume and easy to be absorbed by cells [40,41]. Berberine is more likely to enter the plant and gradually release after loading. Furthermore, CDs have a high adsorption capacity for some cations [42] and can be used as drug carriers. CDs may reduce the strength of berberine adsorption by other ions, increase the content of free berberine, increase the possibility of weed absorption and utilization, and improve herbicidal activity.

### 3.5. Hydroponic Culture Results of Berberine and CDs-Ber

As the herbicidal activity of berberine on barnyard grass was better than on redroot pigweed, in the subsequent experiments, barnyard grass was selected as the target weed for indoor efficacy determination. Figure 6 shows the hydroponic treatment of barnyard grass by berberine and CDs-Ber. Compared with the control group, the CDs had no negative effect on the growth of barnyard grass; after berberine treatment, the barnyard grass seedlings showed obvious chlorosis, and the chlorosis symptoms of the barnyard grass became increasingly serious with the increase in the concentration of active ingredients of berberine. Comparing the damage symptoms of the barnyard grass treated with berberine and CDs-Ber, under the same treatment concentration, the dry and yellowing symptoms of the barnyard grass leaves treated with CDs-Ber were more obvious.

The statistical results of the root length and plant height of the barnyard grass in each group after hydroponic treatment are presented in Table 2. The root length and plant height of the barnyard grass treated with CDs had no significant difference compared with the control group, which indicated that CDs had no inhibitory effect on barnyard grass plants. Compared with the blank control, the growth of barnyard grass was significantly inhibited in all treatments, but there was no significant difference in the growth of barnyard grass between berberine and CDs-Ber at different concentrations, which indicates that the CDs-Ber could not significantly enhance the growth inhibition effect of berberine on mature barnyard grass.

### 3.6. Effects of Berberine and CDs-Ber on Protective Enzymes

As the inhibitory effect of berberine and CDs-Ber on the root length and plant height of barnyard grass was not evident in the above hydroponic method, the changes in the protective enzyme system of barnyard grass were investigated. Furthermore, when the drug concentration was 100 mg/L, the barnyard grass began to yellow; therefore, this concentration was chosen for the following treatment group concentration. The effects of berberine and CDs-Ber on the SOD activity in the roots and leaves of the barnyard grass are shown in Figure 7a. The results show that in the roots and leaves, the SOD activity of the CDs-Ber treatment group was higher than that of the berberine treatment group and control group, indicating that the addition of chemicals could stimulate barnyard grass to increase the expression of SOD, and the stress effect of CDs-Ber on barnyard grass was stronger than that of berberine.

After treatment with berberine and CDs-Ber, the POD activity between the treatment group and the control group is shown in Figure 7b. Overall, both berberine and CDs-Ber enhanced the POD activity. In the roots, the effects of berberine and CDs-Ber on the POD activity were significantly higher than those in the control group, while only the CDs-Ber in the leaves was significantly higher than in the control group, indicating that berberine and CDs-Ber could affect the POD activity of barnyard grass. The stress of CDs-Ber on the leaves was stronger than that of berberine.

From the results presented in Figure 7c show that berberine and CDs-Ber both enhanced the activity of CAT in barnyard grass. The highest enzyme activity was in the 100 mg/L CDs-Ber treatment group, followed by the 100 mg/L berberine treatment group, and the lowest was in the blank control group without exogenous substances. There were significant differences in the enzyme activity between the leaf treatment group and the control group, while there was no difference between the root berberine treatment group and the control group. The results show that CDs-Ber had the greatest influence on the CAT activity of barnyard grass.

SOD, POD, and CAT are the main components of plants’ protective enzymes that can keep active oxygen in plants in a dynamic balance and have endogenous protection and stress resistance [43,44]. Reactive oxygen species (ROS) are a type of small molecule produced during biological aerobic metabolism that can regulate plant growth and development, programmed cell death, and hormone signal transduction [45]. Pesticides boost the activity of SOD-dominated cytoprotective enzymes, removing ROS in plants and effectively preventing oxidative damage [46]. However, when pesticides are used in excess, the protective enzyme system is destroyed, and normal active oxygen metabolism is inhibited, resulting in a continuous increase in the active oxygen content, the acceleration of the membrane lipid peroxidation chain reaction, and the accumulation of peroxide harmful substances, which in turn damages the cell membrane system and macromolecular life substances, causing a series of disorders and eventually leading to plant damage and death [47]. The results of the protective enzyme system show that berberine and CDs-Ber could improve the enzyme activity of SOD, POD, and CAT in barnyard grass, indicating that barnyard grass had produced a large number of protective enzymes to resist damage under the stress of exogenous drugs, which destroyed the balance of the protective enzyme system in barnyard grass, leading to the increase in the ROS content and oxidative damage, and then inhibited the growth of barnyard grass (Figure 1). The activities of SOD, POD and CAT in the roots of the barnyard grass treated with CDs-Ber were significantly higher than those in the control group, and the activities of SOD and CAT were higher than those in the berberine treatment group, while only the activity of CAT in the leaves was significantly higher than that in the control group. It was concluded that the stress effect of CDs-Ber on barnyard grass was stronger than that of berberine, and it also provided a theoretical basis for the experimental results that CDs-Ber had higher herbicidal activity than berberine.

### 3.7. Effects of Berberine and CDs-Ber on the Number of Bacteria in Soil

Pesticides are harmful to soil microbial communities [48]. Killing a specific microbial community, for example, alters the individuals or groups that rely on it, thereby altering the entire soil system [49]. Changes in soil chemistry, pH, and structure may occur as a result of the consumption of soil organic matter [50]. Following the rain, the degraded soil is more vulnerable to surface water erosion, and the soil began to become barren [51]. Therefore, studying the effect of CDs-Ber on soil bacteria and fungi has a certain reference value for safety.

The NA plate and PDA plate, after two days of culture, were taken for the plate count of bacteria and fungi, and the results are shown in Figure 8. The berberine and CDs-Ber treatment groups had a significantly lower number of colonies than the control group. Comparing the number of colonies between the berberine and CDs-Ber treatment groups, there was no significant difference in the antibacterial effect between the two treatments for bacteria or fungi, which indicates, to some extent, that nano-preparation berberine did not have more negative effects on the environment than berberine in the application process. This might be because the CDs-Ber with a nano formulation did not significantly enhance the antibacterial effect. It also demonstrates that CDs-Ber are relatively harmless to soil colonies.

## 4. Conclusions

In this paper, PAA-modified CDs were successfully prepared, and the CDs-Ber nanosystem was constructed by carrying berberine. It was confirmed that CDs-Ber could enhance the inhibitory effect on the growth of barnyard grass and redroot pigweed, to a certain extent, compared with berberine, and the inhibitory effect on barnyard grass was stronger than that of redroot pigweed. CDs-Ber could increase the activity of SOD and CAT in barnyard grass compared with berberine, resulting in an increased ROS content and oxidative damage, and improved weeding activity. This may be due to the nano-pesticides increasing the efficiency of absorption and conduction, although this remains to be explored. In addition, compared with berberine, CDs-Ber had no significant antibacterial effect on the fungi and bacteria and did not have more negative effects on the soil colony environment, which is relatively safe. In conclusion, a new berberine preparation was provided, providing a new choice for the development and application of berberine in green agriculture.

## Data Availability

Not applicable.

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
