# Peer review of "Preparation of Berberine@carbon Dots Nano-Formulation: Synthesis, Characterization and Herbicidal Activity against Echinochloa crus-galli and Amaranthus retroflexus Two Common Species of Weed"

_nanomaterials, 2022, doi:10.3390/nano12244482_

Round 1

Reviewer 1 Report

Review of Nanomaterials manuscript 2038419

Preparation of berberine@carbon dotsnano-formulatio and bioactivity on the weeds

J Xu et al

General comments

This paper seeks to evaluate the activity of carbon nanodots with berberine adsorbed on the surface for the control of two common weed varieties, barnyard grass and redroot pigweed.

A significant amount of work has gone into the study and the authors should be commended for their efforts.

However, I think there are some minor issues that need addressing before I can recommend publication.

After considering my comments I would be happy to review the next submission.

1.       The title needs rethinking – it needs to be more scientific. I suggest the following

Preparation of berberine@carbon nanodots: synthesis, characterisation and herbicidal activity against Echinochloa crus-galli and Amaranthus retroflexus two common species of weed.

It is important that you have the full identification in the title.

I think that the introduction needs more specific information about the two varieties of weed, why they are considered pests and the economic impact of their proliferation. One paragraph about each would be fine, but more information would enable the reader to have an overview of the potential significance of the study. In this you need to mention that Echinochloa crus-galli is one of many different species of barnyard grass and that it is one of the world’s worst weed varieties.. You also need to mention that when uncontrolled it can reduce crop yields and is even capable of interfering with mechanical harvesting.

For redroot pigweed you need to mention that resistance to herbicides has been noted and that it can be deadly to animals when fed in large quantities. Do some research about the negatives of these species and you will allow readers to understand the significance of your research.

Although both these species can be used as food for human consumption when domesticated, there are several negative impacts of uncontrolled growth and proliferation.

Please consider these comments and put some extra information in the introduction.

Other comments

In general some of the practicl methodology needs close examination in order

In section 2.2 it needs some rewriting to make it clearer.

First paragraph would be better as …. CDs were prepared from citric acid, ethylene diamine and polyacrylic acid according to a published method [21]. A solution of 10g CA and 6mL EDA in 100mL deionised water was stirred for 4h at 500rpm at 200°C. 90mL of 10% PAA solution was added and the mixture was stirred for 3h at 80°C.

The next two lines are not clear. I want you to review the practical procedures throughout the document. Make it simple, clear and concise.

2.3 and 2.4 need attention in the same way.

Line 134  I suggest ‘herbicide’, not medicament

2.6 also needs attention

Conclusions

Where are the conclusions? After a paper of this detail I expect probably two paragraphs of conclusions

Figure Captions

The caption for Figure 3 is not clear. The description of Insets makes no sense. They appear to be UV cuvettes, what are they??

I advise you to check through the captions in detail

References

I note that several references need some correction. On the first page of references this applies to

1.       Int. J. Environ. Sci. Tech

4.  This reference needs the species to be itlicised ..Bidens Pilosa

6  Two species names in the title must be italicised.

As a general rule, if it is italicised in the paper title, then you must italicise it in the references in your paper.

Please check carefully through the references.

I commend your efforts and I think we will finish up with a really nice paper from this work. Please resubmit as soon as practical.

Author Response

Thank you very much for your review comments, reply please see word.

Reviewer 2 Report

The authors have investigated the effect of mixtures of berberine and carbon dots (a nanoformulation) on weeds. The results are interesting but some basic characterizatins seems to be missing.

At the end of the introduction the authors state that: "Here, we prepared CDs loaded with berberine nano formulation."

However, some key questins are not answered regarding their formulation.

How much does the presence of CDs enhance the ability to disperse berberine in water?

What is the binding vonstant of berberine to CDs. Or in other words, at equilibrium in water, what is the ratio between berberine bound to CDs and free berberine?

Could the size of berberine@CD aggregates be investigated by e.g. DLS`.

In additon, the auuthors should provide a prober conclusion section.

Some specific comments:

Section 2.2: Please provide more details regarding the preparation of CDs. For example, I guess an autoclave was used?

It is mentioned that the resulting CD solution was extracted three times with acetone. Is this a liquid liquid extraction (as acetone tens to be miscible with water), or what is meant? Please provide a better description.

The use of past tense in the introcustion is awkward. An example: it is written "Therefore, further exploration and research were needed to develop new berberine 43 formulations."

It sounds like such research is not needed any more. I reccomend the authors to change the tense in the introduciton.

Author Response

(The authors gave the same response as above.)
